# A Mobile Application to Help Self-Manage Pain Severity, Anxiety, and Depressive Symptoms in Patients with Fibromyalgia Syndrome: A Pilot Study

**DOI:** 10.3390/ijerph191912026

**Published:** 2022-09-23

**Authors:** Jordi Miró, Meritxell Lleixà-Daga, Rocío de la Vega, Pere Llorens-Vernet, Mark P. Jensen

**Affiliations:** 1Universitat Rovira i Virgili, Carretera de Valls, 43007 Tarragona, Spain; 2Unit for the Study and Treatment of Pain—ALGOS, Research Center for Behavior Assessment (CRAMC), Department of Psychology, Institut d’Investigació Sanitària Pere Virgili, 43007 Tarragona, Spain; 3Faculty of Psychology, Andalucía Tech. Campus de Teatinos, Universidad de Málaga, 29071 Málaga, Spain; 4Biomedical Research Institute of Málaga (IBIMA), Avda. Jorge Luis Borges n°15, Bl.3 Pl.3, 29010 Málaga, Spain; 5Department of Rehabilitation Medicine, University of Washington, Seattle, WA 98195, USA

**Keywords:** fibromyalgia syndrome, mHealth, mobile app, psychological treatment, pain, digital health

## Abstract

Treatment for individuals with fibromyalgia syndrome (FMS) is complex and is not always accessible to those who could benefit. The aim of this study was to conduct a preliminary evaluation of a mobile-app-delivered, cognitive behavioral treatment (CBT)-based intervention in helping adults self-manage fibromyalgia symptoms. A total of 100 adults with FMS (*M [SD]* age = 49.81, *[9.99]* years; 94% women) were given access to the digital treatment program and downloaded the app. Pain severity, anxiety symptoms, depression symptoms, fatigue, and sleep quality were assessed at pre-treatment, post-treatment, and 3-month follow-up. Fifty-three of the potential participants completed the 47-day treatment. Data showed significant improvements in pain severity (*p* = 0.007, *d* = 0.43), anxiety (*p* = 0.011, *d* = 0.40) and depressive symptoms (*p* = 0.001, *d* = 0.50) from pre-treatment to post-treatment. The effect sizes associated with app use are consistent with improvements seen in previously published clinical trials of CBT for FMS. Improvements were generally maintained, although there was some decrease in the outcomes from post-treatment to the 3-month follow-up. Most participants reported that they were very satisfied with the app. The use of the app was associated with similar levels of improvements found with in-person CBT treatment for FMS. Research to evaluate the effectiveness of the app in a controlled trial is warranted.

## 1. Introduction

Fibromyalgia syndrome (FMS), as it is currently defined in the International Classification of Diseases (ICD-11; [1]), is a chronic health condition characterized by chronic widespread pain, fatigue, and sleep problems. Common comorbidities include cognitive, emotional, and somatic symptoms (e.g., memory deficits, depression, dizziness, fatigue and sleep problems; [1,2]). Individuals with FMS report significant interference in their physical, psychological, social, and labor functions [3,4].

Treatment for FMS is complex [5,6]. Recent guidelines have recommended a step-by-step treatment approach, starting with patient education and non-pharmacological approaches [7]. Consistent with these guidelines, non-pharmacological treatments such as cognitive-behavioral therapy result in significant and meaningful improvements in symptoms [7]. Moreover, unlike medications, these treatments do not have negative side effects [8].

Despite the mounting evidence showing that psychosocial treatments for individuals with FMS that teach self-management strategies are effective for reducing pain and improving function [7], there are important barriers that prevent many individuals with FMS from receiving the psychosocial treatments they need. For example, there are few specialized treatment programs for patients with FMS, resulting in long waiting lists for treatment access. Mobile applications could help to address this problem by bringing treatment closer to those who need it when they need it most [9].

Mobile applications have been shown to be effective to help overcome some of these barriers as they can be used by many individuals, at any time and place, and at no cost to users [9,10]. Although there are a large number of health-related mobile apps that target chronic pain, few of these have undergone a thorough validation process, and fewer still have been developed in collaboration with all interested stakeholders or in a way that follows state-of-the-science treatment guidelines [10]. Moreover, none have been developed specifically for individuals with FMS [11].

The objective of this study was to conduct a preliminary evaluation of Fibroline (henceforth referred to as “the app”) for helping adults to self-manage fibromyalgia symptoms, including pain severity, fatigue, sleep, anxiety and depression. Self-reported symptoms are the gold standard for clinical trials [12,13], including those involving patients with FMS [14]. Specifically, we wanted to study changes in the aforementioned symptoms and assess whether participants were satisfied with Fibroline. We hypothesized that participants (1) would be satisfied with the app; and (2) would report significant improvements in in pain severity, fatigue, anxiety symptoms, depression symptoms (all decreases), and sleep quality (an increase), from pre- to post-treatment, which would be maintained at a 3-month follow-up. We also hypothesized that the effect sizes of these changes would be within the ranges of effect sizes reported in clinical trials of psychosocial interventions for fibromyalgia. Finally, we hypothesized that there would be a significant association between the amount of app use and the improvement in the treatment outcome domains.

## 2. Methods

### 2.1. Procedure

Potential participants were recruited through various strategies, including direct contact with diverse FMS patient associations, announcements in social media (Facebook, Twitter and Instagram), and by reaching out to physicians who specialize in the treatment of patients with FMS. This study received ethical approval from the Ethical Committee for Medical Research of the Pere Virgili Health Research Institute (ref.: 2019/04) and was registered (Clinicaltrials.gov Identifier: NCT04942132).

### 2.2. Participants

Interested individuals with a diagnosis of FMS were eligible if they fulfilled the following criteria: (1) were able to speak and read Spanish; (2) were 18 years old or older; (3) owned an Android Operative System-based smartphone; and (4) provided informed consent. A total of 100 individuals showed an interest in participating and were contacted through e-mail and invited to participate in the study. After a brief interview, if found eligible (i.e., if they met inclusion criteria), they could choose to begin participation immediately, or sometime later. Fifty-three of them completed the treatment and responded to outcome measures. Study participation was available from December 2019 to January 2021.

After a participant chose to use the app, a link to the online survey study was sent to them via e-mail. This survey included questionnaires measuring pain intensity, sleep quality, fatigue, anxiety, and depression symptoms. Participants were asked to report every morning and evening for a 7-day period, as the baseline, about their pain intensity, anxiety, depression, fatigue, and sleep quality. This period was also used to identify any problems participants had with using the app.

On the last day of the one-week baseline assessment, participants received an e-mail with the instructions for downloading and installing the app, and a code that allowed to access the app. Once the treatment was over, another e-mail was sent with a link to respond to the post-treatment assessment survey. At the end of the treatment, participants were informed that they could continue using the app if they wished to. Participants did not receive economic compensation for completing the assessments or participating. Study data was collected by research staff.

### 2.3. Measures

*Demographic information*. Participants were asked to describe their sex, age, country of residence, and number of years with a diagnosis of FMS.

*Use of the app*. We recorded the number of reports that the participants uploaded during the treatment period and after that (i.e., post-treatment) using back-end data stored by the app.

*Satisfaction with the**app*. Participants were asked to report their satisfaction with the use of the app at post-treatment on a 0–10 numerical rating scale, where 0 = “Completely dissatisfied” and 10 = “Totally satisfied”.

*Pain severity*. At each assessment-point, participants were asked to rate three domains of pain intensity (i.e., current pain, and average and worst pain in the past 7 days) using 0–10 numerical rating scales (NRS-11) with 0 = “No pain” and 10 = “Severe pain.” We computed a composite score (consisting of the mean of the individuals’ three pain intensity scores) as a measure of pain severity. There is ample evidence that composite scores are adequate for assessing pain intensity in clinical trials (e.g., [15]), and digital versions of the NRS-11 provide valid and reliable data [16].

*Sleep quality.* We used the Pittsburg Sleep Quality Index (PSQI) Spanish version [17] to measure sleep quality. With the PSQI, respondents are asked to answer 19 questions that assess a variety of factors related to sleep quality. In this study, the internal consistency (Cronbach’s alpha) was 0.85.

*Anxiety.* We used the Spanish version of the Patient-Reported Outcomes Measurement Information System short form for Anxiety (PROMIS Short Form v1.0—Anxiety 8a) [18] to assess anxiety. With this measure, respondents are asked to rate the frequency with which they experience 8 anxiety symptoms in the last 7 days, using a 5-point Likert scale (from 1 = ”Never” to 5 = ”Almost always”). In this study, the internal consistency was 0.92. 

*Depression.* We used the Spanish version of the PROMIS short form for Depression (PROMIS Short Form v1.0—Depression 8a) to assess depression symptoms [18]. In this study, the internal consistency was 0.94.

*Fatigue.* We used the Silhouettes Fatigue Scale (SFS; [19]) to assess fatigue. The SFS is a visual scale that presents six human silhouettes representing increasing levels of fatigue, from “No fatigue” to “A lot of fatigue.” The SFS has been shown to be valid and reliable when used with adults with different chronic pain conditions [20,21].

### 2.4. The Mobile Application

Fibroline is a mobile app that has a self-administered cognitive behavioral therapy (CBT)-based psychosocial treatment for individuals with FMS. It was developed by the Chair in Pediatric Pain Universitat Rovira Virgili-Fundación Grünenthal [22]. Although the app was designed to be a resource for individuals with FMS and its use is hypothesized to be associated with improvements in clinical symptoms, it is not intended to be a substitute for professional medical care. The app includes 9 modules that provide information about FMS and its treatment (i.e., General introduction and treatment objectives; Pain; Coping skills; Stress and relaxation; Medication use; Sleep; Physical activity; Thought management; and Relapse prevention. Each module has different resources which are accessible after they are unlocked, including: (1) PDF files that can be read, (2) videos that can be watched, and (3) audios that can be listened to. The information about the topic, the number of pages or the time needed to read or listen to the resources is displayed before the resource is accessed. One of the features of Fibroline is the “Evolution” section, where the user is able to see information from previously recorded data in a graphic form, so progress can be seen for each symptom or variable recorded (see Figure 1). The treatment unfolds over the course of 47 treatment days. Treatment duration is ultimately up to the user, because different modules are unblocked depending on the user’s progress.

### 2.5. Data Analysis

We first computed descriptive statistics for the demographic and study variables. Next, we examined the suitability of the data to ensure that they met the assumptions for the planned analyses. To test the first study hypothesis, we (1) compared pre- and post-treatment scores for all five of the outcome variables (i.e., pain severity, sleep quality, anxiety symptoms, depression symptoms, and fatigue) and (2) post-treatment and follow-up scores, to evaluate changes associated with app access and maintenance of treatment gains, respectively. We also computed the effect sizes associated with these changes (Cohen’s *d*) in order to compare them with changes in these measures observed in clinical trials, which range from 0.01 to 0.17 for improvement in pain severity/intensity [23], 0.00 to 0.18 for improvement in distress (i.e., measures of depression, or anxiety or anxiety/depression) [23], −0.05 to 0.85 for improvement in sleep quality [7], and 0.02 to 0.49 for improvement in fatigue [7], at post-treatment. We planned a priori to use paired t-tests for these comparisons if the variables were normally distributed, and the Wilcoxon t test if they were not normally distributed. To test the study hypothesis about the associations between dose and outcomes, we computed Pearson correlation coefficients between the use of the app and (1) pre- to post-treatment and (2) pre-treatment to follow-up. All data analyses were performed using SPSS v.25 for windows. 

## 3. Results

### 3.1. Description of the Study Sample

All the individuals that showed an interest in participating (100) complied with the inclusion criteria, and they were given access to download the app and participate in the study. However, only 53 individuals participated and completed treatment. Most of them (50; 94%) were women. The sample had a mean age of 50.2 years (SD = 8.4). Most of them lived in Spain (49; 92%) and reported a long history of having FMS (mean number of years with an FMS diagnosis = 8.2; SD = 7.9). No statistically significant differences were in the demographic variables between the participants who initially enrolled and the participants who provided data at each subsequent assessment point. 

### 3.2. Satisfaction with the App

None of the participants reported any problems with either downloading or using the app. In addition, the overwhelming majority of the participants that responded to the question about their satisfaction using the app, reported that they were either highly satisfied (39%; a score of 8–10 on a 0–10 numerical rating scale) or satisfied (52%; a score of 5–7 on a 0–10 numerical rating scale) with the app.

### 3.3. Comparisons between Pre-Treatment, Post-Treatment, and 3-Month Follow-Up Measures of the Outcome Variables

The analyses conducted before the comparisons showed that the variables were not normally distributed. Therefore, we used the Wilcoxon t test for comparisons. Table 1 displays the means and standard deviations of the variables at each assessment period, as well as the results for the comparisons between the assessment periods.

We found statistically significant differences between pre- and post-treatment scores in anxiety, depressive symptoms and pain severity (*p* = 0.007, *p* = 0.004, and *p* = 0.007; respectively. The improvements in depressive symptoms were of a medium effect size (*d* = 0.50), whereas the improvements in pain severity (*d* = 0.43) and anxiety (*d* = 0.40) were small-to-medium. However, all of these effect sizes were either larger or within the range of the effect sizes of the effects of face-to-face cognitive-behavioral treatments for FMS treatment shown in previous research [7]. Moreover, although in this study, we did not find statistically significant pre- and post-treatment changes in sleep quality and fatigue (*d* = −0.06, in both cases), the effect sizes associated with these outcomes were also similar to those shown by previous research. Although all three of the outcomes that evidenced significant pre- to post-treatment improvement then evidenced some worsening (i.e., in the direction of pre-treatment levels) from post-treatment to follow-up, this worsening was statistically significant only for depression symptoms.

Correlations among the use of the app and outcome changes.

The data showed a weak but statistically significant association between the use of the app during treatment and the improvement from pre- to post-treatment in pain severity only (*r* = 0.22, *p* < 0.05).

## 4. Discussion and Conclusions

The key finding of this study is that use of the mobile app was associated with significant immediate improvements across multiple outcomes in a sample of adults with FMS, namely pain severity, anxiety, and depression symptoms. Moreover, the data also showed a statistically significant—although weak—association between the use of the application during treatment and improvements in pain severity. Importantly, the findings showed effect sizes within the ranges reported in previous studies of in-person CBT treatments [7]. In addition, the findings are also consistent with research showing improvements in pain severity in individuals with different chronic pain conditions following an app-delivered CBT-based psychosocial treatment (e.g., [24]).

However, contrary to what was hypothesized, no statistically significant improvements were found for sleep quality and fatigue from pre-treatment to post-treatment. This could be explained by the fact that the sleep content in the app was focused on sleep hygiene, and did not specifically target either severe sleep problems or fatigue. As for sleep quality, the data from previous studies are inconsistent; there are studies using CBT-treatments reporting positive findings and there are also studies showing null findings or small effects [24]. Moreover, it is important to note that although CBT-based treatments for sleep have shown beneficial effects, these benefits appear to be moderate at best [24]. Similarly, in relation to CBT-based treatments for fatigue, the findings from prior research are also inconsistent, with some studies reporting positive findings and others null findings [25,26,27]. Research has also shown that in order for face-to-face CBT treatment to impact fatigue, the approach must be both intensive and extensive [28]. These findings suggest that, for an app to improve sleep and fatigue, the treatment in the app must target these symptoms more specifically, intensively and extensively. Research to examine how modifications in the targets and intensity of apps for addressing different symptoms in fibromyalgia is warranted.

The findings also indicate that the positive effects observed at post-treatment were maintained to some degree at the 3-month follow-up (i.e., outcomes remained better at 3-month than at post-treatment), although they also showed some deterioration with a return towards pre-treatment levels. Maintenance of treatment gains is also a significant issue in face-to-face treatment of individuals with FMS, as a recent systematic review and meta-analysis study has shown [7]. The treatment was developed to facilitate self-management and was expected to work without continued direct professional supervision. However, the findings suggest that this expectation might have been unwarranted. Research to study whether including additional components (or modules), including possibly “booster” modules, might facilitate maintenance of the gains would be useful.

We found that the vast majority of the participants were satisfied with the app, as hypothesized. Although treatment satisfaction has been found to be associated with adherence and improvement in outcome variables in traditional face-to-face treatments (e.g., [29]), the role that treatment satisfaction plays in app use has not yet been studied. Thus, research studying the factors that influence patient satisfaction, as well as the role that treatment satisfaction plays in app adherence, is warranted.

A number of limitations should be considered when interpreting the study findings. First, the study did not include a control condition. Although this was designed and conducted as a pilot study to estimate the effect sizes of improvements in various outcomes, additional studies with control groups are needed to evaluate the efficacy of the app. Second, although the sample size was larger than many other clinical trials testing mobile apps (e.g., [30]), the low sample size can limit power for detecting significant effects, and also may limit the generalizability of study findings. Third, we did not include responder analyses to understand the factors associated with outcome (i.e., to help understand who is most likely to benefit from the app). Additional studies to identify the predictors of treatment response are warranted. Fourth, although all of the enrolled participants were given access to download and use the app, several of them decided not to do so. Unfortunately, the reasons they chose not to download the app and participate is not known to us. In addition, several participants were lost to follow-up. Thus, the findings with respect to representativeness and maintenance of gains may or may not reflect the actual maintenance of gains in the entire sample that were given access to the app. Fifth, we used a 3-month follow-up. Although, 3-month follow-ups are used much more often than longer (e.g., 1-year) follow-ups in FMS clinical trials (cf., [31,32,33]), probably because they are more feasible, future studies with longer follow-up periods are warranted. Sixth, we do not know whether the participants were using other pain treatments during their participation. Although it is highly unlikely that the differences in pre- to post-treatment effects were due only to the use of other concomitant treatments, this possibility cannot be ruled out. Future studies should assess concomitant pain treatment use when possible, and control for this when possible. Seventh, Finally, this preliminary study did not include a condition in which participants were provided with gold-standard treatment (i.e., in-person CBT). Research to address all of these limitations (e.g., a controlled trial with more study participants that includes in-person CBT as a comparison condition) would be an important next step.

Despite this study’s limitations, the findings provide new important information on changes that occur in important outcomes with the use of a mobile application to help adults with FMS self-manage pain severity and psychological function (i.e., anxiety and depression symptoms). This is of particular interest, as this technological alternative can facilitative treatment access to almost anybody and at any time, and help to reduce treatment costs [11,34].

## Figures and Tables

**Figure 1 ijerph-19-12026-f001:**
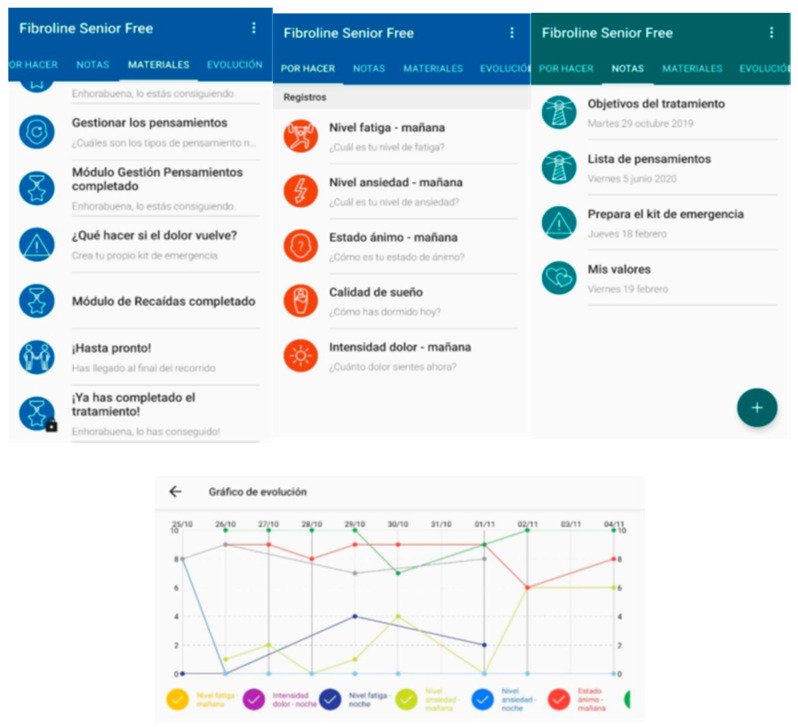
Samples of different screen snapshots from Fibroline.

**Table 1 ijerph-19-12026-t001:** Descriptive statistics and comparisons for the outcome variables.

	Mean Pre-Treatment (SD)	Post-Treatment		Mean 3-Month Follow-Up (SD)		*p* Value Pre-Treatment- Post-Treatment	*p* Value Pre-Treatment- 3-Month Follow-Up	*p* Value Post-Treatment- 3-Month Follow-Up
		*n*	Mean (SD)	*n*	Mean (SD)			
Pain Severity (NRS-11)	7.93 (4.16)	53	7.41 (1.49)	29	7.61 (1.31)	0.007	0.546	0.753
Sleep Quality (PSQI)	13.74 (3.83)	52	14.71 (3.65)	26	13.50 (5.16)	0.874	0.105	0.073
Anxiety (PROMIS)	25.89 (6.81)	52	23.79 (7.61)	23	25.13 (8.122)	0.007	0.081	0.779
Depression (PROMIS)	24.88 (8.09)	48	22.42 (8.39)	19	22.95 (9.02)	0.004	0.868	0.019
Fatigue (SFS)	4.30 (1.19)	48	4.27 (1.25)	19	4.21 (1.36)	0.718	0.856	0.190

Note: NRS = Numerical Rating Scale; PSQI = Pittsburg Sleep Quality Index; PROMIS = Patient-Reported Outcomes Measurement Information System; SFS= Silhouettes Fatigue Scale; SD = Standard Deviation.

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
