# Peer review of "A Mobile Application to Help Self-Manage Pain Severity, Anxiety, and Depressive Symptoms in Patients with Fibromyalgia Syndrome: A Pilot Study"

_ijerph, 2022, doi:10.3390/ijerph191912026_

Round 1
Reviewer 1 Report (New Reviewer)
+ split unjustified lines: 75-92
+ the bioethical commission approval number should also be added at the beginning of the text
+ "Description of the study sample" - move to "Methods"
+ table 1 - change the font
+ the description of the application should be expanded
+ improve the bullet references
Author Response
+ split unjustified lines: 75-92
Authors’ response: Done as requested.
+ the bioethical commission approval number should also be added at the beginning of the text
Authors’ response: Done as requested (see page 2, line 80).
+ "Description of the study sample" - move to "Methods"
Authors’ response: In the Methods section, we provide information about the number of potential participants, and the procedure for their inclusion. In the Results section, we describe the characteristics of the study participants. In that section we also present findings regarding comparisons of the demographic characteristics between the participants initially enrolled and the participants who provided data at each subsequent assessment point. In this revised version, we provide additional information to clarify these issues (see page 5, lines 186-188).
+ table 1 - change the font
Authors’ response: Done as requested.
+ the description of the application should be expanded
Authors’ response: In the revised version, we provide additional information about the application. We also provide several snapshots of screens from the application (see pages 3-4, lines 149-154, and Figure 1).
+ improve the bullet references
Authors’ response: Done as requested.
Reviewer 2 Report (New Reviewer)
Reference 1 does not appear in the text
Line 42. The sentence is too terse and precise more detail
Lines 53 and 54 are not referenced.
In addition to the hypotheses of the study, the objectives of the study should be explicitly stated.
The line spacing is different in the introduction
No red text should appear
Line 96. Why a period of 7 days? Is there literature to support this time?
The variables of the study should be specified in a separate section; if desired within participants, but differentiated from this one.
The authors could have included an image of the app.
The results answer only one of the hypotheses put forward by the authors, user satisfaction with the app.
Reference 21 is very old, aren't there more recent studies?
Author Response
Reference 1 does not appear in the text
Authors’ response: We thank the reviewer for noticing this. We have edited the Introduction and the missing reference has been included (see page 1, line 34).
Line 42. The sentence is too terse and precise more detail
Authors’ response: We wanted to say that treatment for individuals with FMS is complex. We have replaced the word “difficult” with “complex” (see line 39).
Lines 53 and 54 are not referenced.
Authors’ response: In the revised version, we provide a reference supporting our statement (see page 2, line 51).
In addition to the hypotheses of the study, the objectives of the study should be explicitly stated.
Authors’ response: We revised the text to make the study objectives clearer (see page 2, lines 59, and 63-64).
The line spacing is different in the introduction
Authors’ response: The line spacing is now equal throughout the manuscript.
No red text should appear
Authors’ response: Red text was used to make the changes we made easier for the editor and reviewers to see. In the current revision, the only red font is that related to changes made since the previous version.
Line 96. Why a period of 7 days? Is there literature to support this time?
Authors’ response: There is no set number of days used by all studies for a baseline period. However, a baseline period of 7 days is common in health psychology research. We revised the text to make this clearer (see page 2, line 96)
The variables of the study should be specified in a separate section; if desired within participants, but differentiated from this one.
Authors’ response: Done as requested (see page 3, line 106).
The authors could have included an image of the app.
Authors’ response: Thank you for this suggestion. We have now included several snapshots from the app (see Figure 1).
The results answer only one of the hypotheses put forward by the authors, user satisfaction with the app.
Authors’ response: The answer to hypothesis 2 was also included in the text (please see page 5). However, in the previous version, the heading of that section was hidden in the text. We apologize for this formatting error. In the revised version the heading is in italics and can be easily identified (see page 5, lines 201-202).
Reference 21 is very old, aren't there more recent studies?
Authors’ response: There must be some mistake here, as reference 21 is of year 2020.
Reviewer 3 Report (New Reviewer)
Dear Authors,
in my opinion, the topic is interesting considering the emerging role of new technologies for telehealth possibilities in the post-Covid19 era, aiming at reaching the greatest number of patients.
However, I have concerns about the methodological implant of the study and some critical issues should be addressed.
Major revisions
INTRODUCTION. A careful revision is necessary because several punctuation errors should be addressed.
METHODS. Please, report numbers of patient selection process in the “Results” Section. Page 3, line 98. The sentence “None of them reported any problem in either downloading or using the app” should also be reported in the Result Section.
METHODS. Please, clarify exclusion criteria.
METHODS. Please include informed consent obtained between eligibility criteria.
METHODS. Please, clarify how the sample size was calculated.
METHODS. Please characterize the operators who performed the data collection/analysis.
METHODS. Concomitant new treatments during app utilization might bias results. The sample characterization might be significantly improved including data about new treatments started by patients. On the other hand, if this information is not obtainable, it should be discussed in the Limitations Subsection.
RESULTS. Patients assessed for eligibility and patients excluded should be clarified in the results section, characterizing at least the main cause of exclusions. Drop-out reasons should also be clarified.
DISCUSSION. This Section should be improved, citing the importance of telehealth development, especially focusing on the possibility of reaching patients that normally would not benefit from clinical care. You could mention some clinical application of new technologies for telehealth. According to this, you should cite the following references:
· Invernizzi, M., et al (2020). Integrating Augmented Reality Tools in Breast Cancer Related Lymphedema Prognostication and Diagnosis. Journal of visualized experiments : JoVE, (156), 10.3791/60093. https://doi.org/10.3791/60093
· Galway K et al. Adapting Digital Social Prescribing for Suicide Bereavement Support: The Findings of a Consultation Exercise to Explore the Acceptability of Implementing Digital Social Prescribing within an Existing Postvention Service. Int J Environ Res Public Health. 2019 Nov 18;16(22):4561. doi: 10.3390/ijerph16224561. P
· Lippi, L., et al (2022). Closing the Gap between Inpatient and Outpatient Settings: Integrating Pulmonary Rehabilitation and Technological Advances in the Comprehensive Management of Frail Patients. International journal of environmental research and public health, 19(15), 9150. https://doi.org/10.3390/ijerph19159150
Author Response
Dear Authors,
in my opinion, the topic is interesting considering the emerging role of new technologies for telehealth possibilities in the post-Covid19 era, aiming at reaching the greatest number of patients.
Authors’ response: We thank the reviewer for the kind comments.
However, I have concerns about the methodological implant of the study and some critical issues should be addressed.
Major revisions
INTRODUCTION. A careful revision is necessary because several punctuation errors should be addressed.
Authors’ response: We thank the reviewer for noticing these errors, and apologize for them. We have revised the Introduction to address those punctuations errors which have been now corrected. In addition, we have edited the Introduction for clarification.
METHODS. Please, report numbers of patient selection process in the “Results” Section. Page 3, line 98.
Authors’ response: There must be some sort of mistake in the page/line alluded as the information in page 3, line 98 is not related to the number of participants. Nevertheless, in the Methods section, we provide information about the number of potential participants, and the procedure for their inclusion. In the Results section, we provide information regarding the characteristics of study participants. In that section we also present findings regarding comparisons between the individuals who initially enrolled and those participants who ultimately provided data at each assessment point in relation to demographic variables. In this revised version, we provide additional information to clarify this issue (see page 5, lines 186-188).
The sentence “None of them reported any problem in either downloading or using the app” should also be reported in the Result Section.
Authors’ response: Thank you for this suggestion, and we agree that this information belongs to the Results section. Therefore, in order to avoid repetition we have moved this sentence to the Results section, and placed at the beginning of the “Satisfaction with the app” heading (see page 5, line 195-196).
METHODS. Please, clarify exclusion criteria.
Authors’ response: We did not have any exclusion criteria. Those participants who did not meet the inclusion criteria were not included.
METHODS. Please include informed consent obtained between eligibility criteria.
Authors’ response: Done as requested (see page 2, line 86).
METHODS. Please, clarify how the sample size was calculated.
Authors’ response: This was a pilot study. There was therefore not a sample calculation. However, we provided information on the effect size of the differences found, as described in the Data analysis and reported in the Results section (see page 5; lines 173-178 and lines 207-216).
METHODS. Please characterize the operators who performed the data collection/analysis.
Authors’ response: We revised the text to make it clear who collected the study data (these were research staff, see page 3, line 105). The person who conducted the data analysis was Meritxell Lleixà-Daga, who is a co-author on the paper. However, we do not usually (in fact, among the over 800 papers published by the authors over the years, we have never) explicitly noted who conducted the data analysis in a paper, so we are not sure what purpose this would serve here. If the editor or reviewer believe this is important, we would be happy to call out the co-author who conducted the data analyses in the Results section. Please let us know if you think this would improve the paper.
METHODS. Concomitant new treatments during app utilization might bias results. The sample characterization might be significantly improved including data about new treatments started by patients. On the other hand, if this information is not obtainable, it should be discussed in the Limitations Subsection.
Authors’ response: The reviewer raises an important point about a potential biasing factor. Unfortunately, information about concomitant pain treatments was not collected. To address this issue, we now note this as a study limitation (see page 8, line 289-294).
RESULTS. Patients assessed for eligibility and patients excluded should be clarified in the results section, characterizing at least the main cause of exclusions. Drop-out reasons should also be clarified.
Authors’ response: No potential participants were excluded from the study; all that showed an interest and were willing to participate and who met inclusion criteria were included. However, although all (100) of the enrolled participants were given access to download and use the app, several of them decided not to do so. Unfortunately, the reasons they chose not to download the app is not known to us. We have included this information as a limitation in the revised manuscript (see page 7, lines 281-285).
DISCUSSION. This Section should be improved, citing the importance of telehealth development, especially focusing on the possibility of reaching patients that normally would not benefit from clinical care. You could mention some clinical application of new technologies for telehealth. According to this, you should cite the following references:
- Invernizzi, M., et al (2020). Integrating Augmented Reality Tools in Breast Cancer Related Lymphedema Prognostication and Diagnosis. Journal of visualized experiments : JoVE, (156), 10.3791/60093. https://doi.org/10.3791/60093
- Galway K et al. Adapting Digital Social Prescribing for Suicide Bereavement Support: The Findings of a Consultation Exercise to Explore the Acceptability of Implementing Digital Social Prescribing within an Existing Postvention Service. Int J Environ Res Public Health. 2019 Nov 18;16(22):4561. doi: 10.3390/ijerph16224561. P
- Lippi, L., et al (2022). Closing the Gap between Inpatient and Outpatient Settings: Integrating Pulmonary Rehabilitation and Technological Advances in the Comprehensive Management of Frail Patients. International journal of environmental research and public health, 19(15), 9150. https://doi.org/10.3390/ijerph19159150
Authors’ response: Thank you for this suggestion. We added text to note the importance of mobile applications in health care, and provide citations supporting this statement (see page 2, lines 52-54). In addition, we added text to the Discussion and provide two citations that are specifically related to mobile applications and chronic pain-related conditions, as this is the topic of our study (see page 7, line 304).
Reviewer 4 Report (New Reviewer)
Summary
This manuscript studies the effectiveness of using the mobile app in a sample of adults with FMS across multiple outcomes such as anxiety and depression. While this problem is fundamentally important, there are critical issues that prevent it from being accepted such as the inconsistency between the displayed results and the computed values in the analysis, as well as the conflicting conclusion given by the results.
Major comments:
1. Overall, the results section needs improvement. Specifically, the data displayed in Table 1 does not seem to align with the analysis given in the main body. For example, in lines 189 - 190, the p-value of anxiety and depression are inconsistent with the table. Also, the sentence in lines 183-184 seems incomplete.
2. The result analysis leverages cohen's d value to measure the difference between two groups. However, the given d values do not seem to be computed from the data shown in Table 1. For example, the d value of depressive symptoms between pre- and post-treatment using the data in Table 1 seems to be 0.33, rather than the value given in lines 191-192 as the main finding. It is recommended to provide the equation used to compute d value in the manuscript for clarity.
3. Following the above comments, the claim about the maintenance of positive effects observed at post-treatment and 3-month follow-up is questionable. Specifically, the Sleep Quality value does not follow the pattern of maintenance (it drops below the pre-treatment). It is recommended to provide further analysis and in-depth explanation.
4. As indicated in the manuscript as one of the limitations, the participants are heavily biased in terms of gender. Also, the impact of lacking follow-up participants is significant (e.g., n drops from 48 to 19 for Depression and Fatigue). These issues make it hard to justify the conclusion given in the manuscript.
Minor comments:
The text format first paragraph is problematic with inconsistent parenthesis.
Author Response
This manuscript studies the effectiveness of using the mobile app in a sample of adults with FMS across multiple outcomes such as anxiety and depression. While this problem is fundamentally important, there are critical issues that prevent it from being accepted such as the inconsistency between the displayed results and the computed values in the analysis, as well as the conflicting conclusion given by the results.
Major comments:
- Overall, the results section needs improvement. Specifically, the data displayed in Table 1 does not seem to align with the analysis given in the main body. For example, in lines 189 - 190, the p-value of anxiety and depression are inconsistent with the table. Also, the sentence in lines 183-184 seems incomplete.
Authors’ response: Thank you for noting the typographical errors. We revised the paper to correct them (see Table 1).
- The result analysis leverages Cohen's d value to measure the difference between two groups. However, the given d values do not seem to be computed from the data shown in Table 1. For example, the d value of depressive symptoms between pre- and post-treatment using the data in Table 1 seems to be 0.33, rather than the value given in lines 191-192 as the main finding. It is recommended to provide the equation used to compute d value in the manuscript for clarity.
Authors’ response: As mentioned in our previous response, we have revised and corrected the information provided. We used JASP for this specific analysis. The equation, as it is shown in the official web page (https://jasp-stats.org/) is the following:
- Following the above comments, the claim about the maintenance of positive effects observed at post-treatment and 3-month follow-up is questionable. Specifically, the Sleep Quality value does not follow the pattern of maintenance (it drops below the pre-treatment). It is recommended to provide further analysis and in-depth explanation.
Authors’ response: We revised the text in the Discussion, to note the key findings indicating immediate improvements in three of the outcomes: pain severity, anxiety and depression symptoms, and -contrary to what was hypothesized- no effects on sleep quality (see page 6, lines 231-241). We also provide potential explanations for the findings that were inconsistent with our hypothesis (see page 6, lines 241-255).
- As indicated in the manuscript as one of the limitations, the participants are heavily biased in terms of gender.
Authors’ response: The author is correct that the majority of the participants were women, consistent with FMS being more prevalent in women than in men, with women comprising about 85% to 95% of the total FMS patient population across clinical studies (for example, see Clauw D.J. Fibromyalgia: a clinical review. JAMA. 2014;311(15):1547–1555). For this reason, research in FMS has primarily focused on women, as it is the case of our study. The fact is that the sample composition reflects the population of individuals with FMS.
Also, the impact of lacking follow-up participants is significant (e.g., n drops from 48 to 19 for Depression and Fatigue). These issues make it hard to justify the conclusion given in the manuscript.
Authors’ response: We agree with the reviewer on these points. We note the high dropout rate as a significant study limitation, making any conclusions regarding longer-term effects tentative, at best (see page 7, lines 284-286).
Minor comments:
The text format first paragraph is problematic with inconsistent parenthesis.
Authors’ response: Thank you for noting this inconsistency. We corrected it in the revision.
Round 2
Reviewer 2 Report (New Reviewer)
The text has been substantially improved and I have no objection to its acceptance.
Author Response
The text has been substantially improved and I have no objection to its acceptance.
Authors’ response: We thank the reviewer for the kind comments.
Reviewer 3 Report (New Reviewer)
Dear Authors,
thank you for your careful revision of the manuscript. I think it has been significantly improved during the revision process. However, the reference list is still poor. You could still improve the discussion section by highlighting the role of digital innovation in clinical settings and the increasing interest in this field after the COVID pandemic. You should cite the following reference published in this Journal.
-Galway K et al. Adapting Digital Social Prescribing for Suicide Bereavement Support: The Findings of a Consultation Exercise to Explore the Acceptability of Implementing Digital Social Prescribing within an Existing Postvention Service. Int J Environ Res Public Health. 2019 Nov 18;16(22):4561. doi: 10.3390/ijerph16224561. P
- Lippi, L., et al (2022). Closing the Gap between Inpatient and Outpatient Settings: Integrating Pulmonary Rehabilitation and Technological Advances in the Comprehensive Management of Frail Patients. International journal of environmental research and public health, 19(15), 9150. https://doi.org/10.3390/ijerph19159150
Author Response
Dear Authors,
thank you for your careful revision of the manuscript. I think it has been significantly improved during the revision process.
Authors’ response: We thank the reviewer for the kind comments.
However, the reference list is still poor. You could still improve the discussion section by highlighting the role of digital innovation in clinical settings and the increasing interest in this field after the COVID pandemic. You should cite the following reference published in this Journal.
-Galway K et al. Adapting Digital Social Prescribing for Suicide Bereavement Support: The Findings of a Consultation Exercise to Explore the Acceptability of Implementing Digital Social Prescribing within an Existing Postvention Service. Int J Environ Res Public Health. 2019 Nov 18;16(22):4561. doi: 10.3390/ijerph16224561. P
- Lippi, L., et al (2022). Closing the Gap between Inpatient and Outpatient Settings: Integrating Pulmonary Rehabilitation and Technological Advances in the Comprehensive Management of Frail Patients. International journal of environmental research and public health, 19(15), 9150. https://doi.org/10.3390/ijerph19159150
Authors’ response: Thank you for this suggestion. In relation to this request, in our previous review, we added text to note the importance of mobile applications in health care, and provide citations supporting this statement (see page 2, lines 52-54). In addition, we added text to the Discussion and provided two citations that are specifically related to mobile applications and chronic pain-related conditions, as this is the topic of our study (see page 7, line 304). We appreciate the suggestion of including these two interesting studies published in this journal. We carefully read them and found that although of interest they are not related to our study at all. Therefore, instead of diverting the attention of future readers to an area that is not in the aims of our study, we prefer to keep focused in the area of mobile health -which is the specific area and objective of our study-, and cite studies that are, in fact, related to the area of the study.
Reviewer 4 Report (New Reviewer)
Summary
The authors have attempted to improve the paper based on comments from the previous round. However, there is one issue remains unaddressed as follows.
Major comment 2: The equation to compute cohen's d is not given in the author's reply or the revised manuscript, thus it is still hard to justify the analysis results.
Author Response
The authors have attempted to improve the paper based on comments from the previous round. However, there is one issue remains unaddressed as follows.
Major comment 2: The equation to compute cohen's d is not given in the author's reply or the revised manuscript, thus it is still hard to justify the analysis results.
Authors’ response: We apologize for this. However, in our response to this specific request, we provided the equation. It is unclear why this information did not show in the response that we uploaded. Maybe, this could be related to the fact that it was an image, and the system did not support it. This is our best hypothesis, and we believe this could be the case because in the letter to the Editor, which was uploaded as a pdf (this letter is a copy of our responses to all the reviewers’ suggestions/comments), the equation is there. We apologize for the inconveniences this have caused. In order to make secure that the equation it is now shown, we are uploading our response to this comment as a pdf.
As mentioned in our previous response, we have revised and corrected the information provided. We used JASP for this specific analysis. The equation, as it is shown in the official web page (https://jasp-stats.org/) is the following:
Again, the image is not uploaded, and I the system is not allowing to upload a pdf. It is unclear why. I am asking the Editors to share this information with the reviewer.

This manuscript is a resubmission of an earlier submission. The following is a list of the peer review reports and author responses from that submission.
Round 1
Reviewer 1 Report
Dear Authors,
The present study focuses on developing the application Fibroline® for patient assessment. In the study, the application of Fibroline® should not be reported as a therapeutic or clinical procedure. The application outcomes analysis does not focus on the patient's clinical status but their perception of symptoms. The clinical examination by the health professional should be recorded before and after the application use, in the flow-up procedure, and following a standardized therapeutic. In the absence of this monitorization, the methodology presented is not adjusted for the aim of the study.
Several suggestions can be added, such as "Long waiting lists" should be discussed regarding health systems equity (lines 34-35). It is a quality indicator of the health organization, and it should be considered in the global healthcare, and not for the focus proposed, the fibromyalgia. In addition, the authors should clarify the issue "Recently, mobile applications have been shown to be effective to help overcome some of these barriers" (lines 37-38). A follow-up f three months is not enough; it should be one year, the minimum. The lines 244-243 and the lines 251-252 are repeated.
Author Response
The present study focuses on developing the application Fibroline for patient assessment. In the study, the application of Fibroline should not be reported as a therapeutic or clinical procedure.
Authors’ response: Thank you for this suggestion. We reviewed the paper and edited it carefully to ensure that the app was not described as a clinical procedure or intervention; instead, we emphasize that the app is a resource that patients with FMS can use to help self-manage FMS symptoms (see changes on pages 3 lines 1351-138 where the app is being described).
The application outcomes analysis does not focus on the patient's clinical status but their perception of symptoms.
Authors’ response: The reviewer is correct that the primary outcomes in this study are self-reported FMS symptoms, consistent with the goals of the app (i.e., to help individuals with FMS manage symptoms). We reviewed the paper carefully to ensure that there was no language implying that the goal of the app is to impact clinical status. In addition, we added text to emphasize the fact that patient self-reported symptoms are the gold standard for clinical trials (see pages 2, lines 61-63).
The clinical examination by the health professional should be recorded before and after the application use, in the flow-up procedure, and following a standardized therapeutic. In the absence of this monitorization, the methodology presented is not adjusted for the aim of the study.
Authors’ response: Including a full clinical evaluation by a health care professional before and after app use is an interesting idea, although we are not aware of any clinical trial that includes this as part of their procedures (see, for example, two clinical trials published in the JAMA: (1) Pandrangi VC, Shah SN, Bruening JD, Wax MK, Clayburgh D, Andersen PE, Li RJ. Effect of Virtual Reality on Pain Management and Opioid Use Among Hospitalized Patients After Head and Neck Surgery: A Randomized Clinical Trial. JAMA Otolaryngol Head Neck Surg. 2022 Jun 9:e221121. doi: 10.1001/jamaoto.2022.1121; (2) Garland EL, Hanley AW, Nakamura Y, Barrett JW, Baker AK, Reese SE, Riquino MR, Froeliger B, Donaldson GW. Mindfulness-Oriented Recovery Enhancement vs Supportive Group Therapy for Co-occurring Opioid Misuse and Chronic Pain in Primary Care: A Randomized Clinical Trial. JAMA Intern Med. 2022 Apr 1;182(4):407-417. doi: 10.1001/jamainternmed.2022.0033.). Although such as design feature would be unusual in a clinical trial, we agree with the reviewer that including a physical evaluation before and after the app use would provide an additional perspective on outcomes. We note this idea in the limitations section of the revised paper (see page 6; lines 250-254).
Several suggestions can be added, such as "Long waiting lists" should be discussed regarding health systems equity (lines 34-35). It is a quality indicator of the health organization, and it should be considered in the global healthcare, and not for the focus proposed, the fibromyalgia.
Authors’ response: We agree with the reviewer that the existence of long waiting lists is a problem with health care quality and equity, and is not a problem limited to FMS alone. On the other hand, it is a problem that could be addressed by the app for individuals with FMS who have to deal with this issue. We revised the text to make these points clearer (see page 2; lines 48-51).
In addition, the authors should clarify the issue "Recently, mobile applications have been shown to be effective to help overcome some of these barriers" (lines 37-38).
Authors’ response: Thank you for this suggestion. We have edited the text to provide the suggested clarification on this issue (see page 2, lines 53-54).
A follow-up f three months is not enough; it should be one year, the minimum.
Authors’ response: We agree with the reviewer that a one year (or even longer) follow-up provides a more rigorous test of the maintenance of benefits than a 3-month follow-up does. That said, 3-month follow-ups are used much more often than 1 year or longer follow-ups in FMS clinical trials (cf., Carson JW, Carson KM, Jones KD, Mist SD, Bennett RM. Follow-up of yoga of awareness for fibromyalgia: results at 3 months and replication in the wait-list group. Clin J Pain. 2012 Nov-Dec;28(9):804-13. doi: 10.1097/AJP.0b013e31824549b5.; Serrat M, Sanabria-Mazo JP, Almirall M, Musté M, Feliu-Soler A, Méndez-Ulrich JL, Sanz A, Luciano JV. Effectiveness of a Multicomponent Treatment Based on Pain Neuroscience Education, Therapeutic Exercise, Cognitive Behavioral Therapy, and Mindfulness in Patients With Fibromyalgia (FIBROWALK Study): A Randomized Controlled Trial. Phys Ther. 2021 Dec 1;101(12):pzab200. doi: 10.1093/ptj/pzab200; Schweiger V, Secchettin E, Castellani C, Martini A, Mazzocchi E, Picelli A, Polati E, Donadello K, Valenti MT, Dalle Carbonare L. Comparison between Acupuncture and Nutraceutical Treatment with Migratens® in Patients with Fibromyalgia Syndrome: A Prospective Randomized Clinical Trial. Nutrients. 2020 Mar 19;12(3):821. doi: 10.3390/nu12030821; Udina-Cortés C, Fernández-Carnero J, Romano AA, Cuenca-Zaldívar JN, Villafañe JH, Castro-Marrero J, Alguacil-Diego IM. Effects of neuro-adaptive electrostimulation therapy on pain and disability in fibromyalgia: A prospective, randomized, double-blind study. Medicine (Baltimore). 2020 Dec 18;99(51):e23785. doi: 10.1097/MD.0000000000023785) probably because they are more feasible. We revised the limitations paragraph to discussion this study limitation (see page 7, lines 262-265).
The lines 244-243 and the lines 251-252 are repeated.
Authors’ response: Thank you for noting this typographical error. The repeated text was removed from the text.
Reviewer 2 Report
The introduction of this document needs to be improved. It does not speak at any time of the Fibromyalgia Syndrome, it does not indicate its main clinical aspects nor does it indicate the treatments that are being carried out on this pathology.
They must include what their diagnosis is based on, who is affected, the level of education of these people to know if they can interact properly with the devices that they indicate.
How was the sample recruited? Were they searched by phone or some database?
Why was it decided that the sample would be 100? No sample calculation done?
To carry out a quality pilot study, a control group would be necessary so that the results could be comparable and extrapolated.
Author Response
Reviewer 2
The introduction of this document needs to be improved. It does not speak at any time of the Fibromyalgia Syndrome, it does not indicate its main clinical aspects nor does it indicate the treatments that are being carried out on this pathology.
Authors’ response: Thank you for this suggestion to improve the paper further. We revised the text in the Introduction to provide additional information about FMS, its key clinical aspects, and its treatments.
They must include what their diagnosis is based on, who is affected, the level of education of these people to know if they can interact properly with the devices that they indicate.
Authors’ response: Thank you for these suggestions. We revised the text to clarify how the diagnosis of FMS was confirmed (see page 3, lines 80-9884 We also added details regarding how well participants were able to use the app (see page 3, lines 92-94). We did not include responder analyses to understand the factors associated with outcome (i.e., to help understand who is most affected by the app), because (1) this was not an aim of the study and (2) more importantly, the sample size is not large enough to have the power to conduct such analyses. To address this third issue, we note the lack of these analyses to predict treatment responders as a study limitation, and recommend that future studies with larger sample sizes conduct such analyses (see page 6, lines 257-260).
How was the sample recruited? Were they searched by phone or some database?
Authors’ response: We agree with the reviewer that the paper would be stronger with additional information regarding the specific strategies used for subject recruitment. We revised the text to add this information (see page 2, lines 73-77).
Why was it decided that the sample would be 100? No sample calculation done?
Authors’ response: As this was a pilot study, the goal was to include enough subjects to be able to make reliable estimates of changes that occur with app use. Thus, the focus of the study was on the effect sizes of the differences found in outcomes. We revised the text to make this clearer (see page 4, lines 154-159 and lines 184-192).
To carry out a quality pilot study, a control group would be necessary so that the results could be comparable and extrapolated.
Authors’ response: We concur with the reviewer on this point. The lack of a control condition is an important study limitation. We revised the text to make this limitation clearer, and discussed the need for future research with the app to include a control condition(see page 6 lines 248-250).
Round 2
Reviewer 1 Report
Dear Authors,
The International Classification of Diseases, of the World Health Organization, is a classification tool for disease codification as a standardized procedure for a database. It is not a reference for clinical diagnostic criteria or disease definition. The authors can record the definition of the American College of Rheumatology (https://www.rheumatology.org/Portals/0/Files/2010_Preliminary_Diagnostic_Criteria.pdf). In this sense, it can focus on the widespread pain index (WPI).
The authors made a relevant effort to improve the manuscript. However, my opinion stands on its rejection. Its development has profound flaws, namely in the study design, the selection of patients, and its follow-up. The present study's clinical trial design is not the study design. The authors focus on several references to clinical trials with the control group and its absence in the present design study.
The patient selection (potential patient) is not following inclusion and exclusion criteria. The "brief interview" for eligible patients should be explain. The number of the sample is incorrect (53? no 100). The monitorization of follow-up is time-short, and it do not allow a comparison of data, namely in the absence of a control group.
Reviewer 2 Report
ok